# A Novel Approach for Organic Strawberry Cultivation: Vermicompost-Based Fertilization and Microbial Complementary Nutrition

Neslihan Kilic [1], Hayriye Yildiz Dasgan [2],* and Nazim S. Gruda [3]

1 Department of Organic Farming Management, Kadirli Faculty of Applied Sciences, University of Osmaniye Korkut Ata, Osmaniye 80000, Turkey; neslihankilic@osmaniye.edu.tr
2 Department of Horticulture, Faculty of Agriculture, University of Cukurova, Adana 01330, Turkey
3 Institute of Plant Sciences and Resource Conservation, Division of Horticultural Sciences, University of Bonn, D-53113 Bonn, Germany; nazim.gruda@hu-berlin.de
* Correspondence: dasgan@cu.edu.tr

**Abstract:** This study investigated the effects of vermicompost fertilization with complementary microbial nutrition on the plant growth, yield, and fruit quality of the organically grown strawberry "Monterey" cultivar. Along with vermicompost, five different microbial fertilizers containing plant-growth-promoting rhizobacteria (PGPR) and arbuscular mycorrhizal fungi (AMF) were used as complementary nutrition. Here, we examined plant growth parameters, strawberry yield, fruit weight, pH, total soluble solids, and acidity in fruit and leaf mineral nutrient concentrations. Vermicompost-based fertilization with PGPR and AMF improved plant growth, yield, and fruit quality. The highest total yield (216.75 g per plant$^{-1}$) and heaviest fruits with an average of 18.11 g were obtained from the vermicompost-based fertilization with PGPR containing complementary fertilization. This included *Bacillus amyloliquefaciens*, *Bacillus pumilus*, *Bacillus subtilis*, *Bacillus licheniformis*, *Bacillus megaterium*, *Trichoderma harzianum*, and *Trichoderma konigii*. This treatment also resulted in the best ratio of total soluble solids to acidity (18.74), pH (3.95), and mineral nutrient concentrations in leaves. The novel approach with vermicompost-based fertilization and complementary microbial nutrition improves organic strawberries' growth, yield, and fruit quality. These results are promising for enhancing organic strawberry production.

**Keywords:** *Fragaria* × *ananassa* Duch.; vermicompost; bacteria; mycorrhiza; yield; fruit quality





## 1. Introduction

The strawberry (*Fragaria* × *ananassa* Duch.), which can be grown in different ecological conditions, is one of the most popular berry fruits globally. The strawberry is an excellent source of natural antioxidants, such as carotenoids, phenolics, vitamins, anthocyanins, and flavonoids. It has a very high capacity to scavenge free radicals that can prevent certain types of cancer, cardiovascular diseases, inflammation, obesity, and type II diabetes. Additionally, strawberries are a good vitamin C source and are in high demand by consumers due to their sensory and nutritional properties [1–3]. The strawberry is available for fresh consumption, processed, or frozen. The recoup of strawberry investments in the first year, especially when other fruits are unavailable in the market, provides a decent source of income for the producers [4].

The plantation area and production of strawberries have increased substantially in the last five years. According to FAO data, 377,112 ha of worldwide strawberry production area in 2015 expanded to 389,665 ha in 2021. Meanwhile, the annual production amount of 8,221,263 tons in 2015 increased to 9,175,384 tons in 2021. Among all the countries producing strawberries, Turkey ranks third worldwide with 669,195 tons of strawberry production [5]. However, organic strawberry production is very limited. The excessive

utilization of chemical fertilizers and pesticides has contributed to soil quality and ecological stability degradation, leading to human health concerns. As a response, there has been a growing trend towards organic farming practices and a corresponding increase in demand for organic fruit products. Pesticide use is intense during conventional strawberry cultivation. Producing organic strawberries is valuable as it provides a safe, chemical-free, and health-protective food source that is particularly appealing to children. However, one of the significant challenges in organic farming is ensuring adequate plant nutrition. As a result, numerous studies have been conducted to enhance the efficacy of various organic sources, such as farm manure, vermicompost, and different biofertilizers for use in organic strawberry cultivation [6–9].

Consumers have shown increasing interest in the crop's nutritional quality [10–12]. According to Dasgan et al. [13–15], biofertilizers could improve crops' antioxidant, vitamin, and mineral contents.

In recent years, the use of vermicompost has increased in horticulture as a growing substrate component [16,17] or as a fertilizer with an increasing momentum [18–22]. It has been acknowledged for its elevated levels of NPK, humic contents, micronutrients, actinomycetes, and beneficial soil microbes, which enhance soil's physical, chemical, and biological characteristics. Earthworms facilitate the biodegradation of organic waste materials through enzymatic action and decomposer microorganisms. In addition to enhancing the composting process, earthworms disinfect and detoxify the product [18]. The resulting improvements in plant growth, development, and productivity lead to a rise in strawberry yield, alongside essential elements, vitamins, and enzymes attributable to vermicompost application [19–22].

In addition, biofertilizers, such as plant-growth-promoting rhizobacteria (PGPR) and arbuscular mycorrhizal fungi (AMF), have gained popularity in horticulture due to their positive impact on plant nutrition [13–15]. These biofertilizers promote plant growth through various mechanisms, including nitrogen fixation; phosphorus solubilization; the production of plant hormones, such as indole acetic acid and cytokinins; and the promotion of gibberellin synthesis within the plant [23,24]. They are also reported to provide organic acids and enzymes, such as 1-aminocyclopropane-1-carboxylate-deaminase, chitinase, and glucanase, to increase water and nutrient uptakes and enhance systemic resistance against diseases [25–28].

The novelty of this study lies in the utilization of "vermicompost-based fertilization and microbial complementary (top-dress) nutrition" as an organic approach to plant nutrition. In organic strawberry cultivation, using mineral chemical synthetic fertilizers is strictly prohibited. Consequently, alternative methods must be employed to enhance plant nutrition. To ensure accessibility for strawberry farmers, PGPR and AMF biofertilizers were chosen from commercially available preparations. While previous studies have explored the individual use of vermicompost, PGPR, and AMF, these investigations were not explicitly focused on organic farming. Our study, however, synergistically combines vermicompost and microbial fertilizers to determine the most suitable combination for organic strawberry nutrition.

This study was carried out to improve plant nutrition in organic strawberry production to increase yields and enrich product quality. We aimed to investigate combinations of different strains of bacteria and mycorrhizae. It was investigated whether vermicompost can promote organic strawberry production alone or in combination with PGPR and AMF. We investigated the effects of vermicompost as a basic fertilizer mixed into the soil in combination with top fertilization by PGPR and AMF treatments on the "Monterey" strawberry variety.

## 2. Material and Methods

### 2.1. Trial Area, Plant Material, and Treatments

This study was carried out in the experimental area of Osmaniye Korkut Ata University in 2019–2020. This region has a Mediterranean climate, and its height from the sea is 121 m.

A medium-neutral day variety, the "Monterey", was used as a strawberry plant material with outstanding flavour and a distinct sweet taste. The fruits are large, firm, slightly carved, and dark red. "Monterey" fruits are used for both fresh markets and processing. The variety is sensitive to mildew, and its earliness and plant structure are comparatively strong. The strawberry crop phenotypic characteristics can vary according to growing and environmental conditions [22].

The vermicompost used in the study, which was produced by a company called Ekosol, was of two types: The first is in a granular form with the name of Ekosolfarm®. It is 100% organic vermicompost obtained from the red California culture worms. It contains 35% organic matter, 1.2% total nitrogen (N), 1% organic nitrogen, and 20% total humic and fulvic acids. The second, a liquid vermicompost fertilizer, contains 7% organic matter, 1% total nitrogen, and 6% total humic fulvic acid (https://www.ekosol. net/urunler_menu, accessed on 5 January 2023). The granular vermicompost as a basic fertilizer was mixed into the soil before seedling transplanting. Liquid vermicompost with the same trade name, Ekosolfarm®, was also applied every week periodically with drip irrigation. Moreover, we examined several biofertilizers as top fertilizers containing PGPR and AMF microorganisms. The biofertilizer with the trade name RhizoFill® included three PGPR species: *Bacillus subtilis*, *Bacillus megaterium*, and *Pseudomonas fluorescens*, at a concentration of $1 \times 109$ cfu/mL. The biofertilizer Subtima® included only *Bacillus subtilis* at a concentration of $1 \times 109$ cfu/mL. The biofertilizer Fontera microzone® contained nine PGPR species at a concentration of $5 \times 108$ cfu/g, including *Azosprillium brasilense*, *Azotobacter vinelandii*, *Rhizobium trifollii*, *Pseudomonas fluorescens*, *Bacillus subtilis*, *Bacillus licheniformis*, *Azotobacter chroococcum*, *Bacillus amyloliquefaciens*, and *Bacillus mucilaginosus*. The biofertilizer Endo Roots Soluble (ERS)® contained nine AMF species at a concentration of $1 \times 104$ cfu/g, including *Glomus intraradices*, *Glomus aggregatum*, *Glomus mosseae*, *Glomus clarum*, *Glomus monosporum*, *Glomus deserticola*, *Glomus brasilianum*, *Glomus etunicatum*, and *Gigaspora margarita*. Finally, the biofertilizer Bontera® contained five PGPR and two AMF species at a concentration of $1.2 \times 108$ cfu/mL, including *Bacillus amyloliquefaciens*, *Bacillus pumilus*, *Bacillus subtilis*, *Bacillus licheniformis*, *Bacillus megaterium*, *Trichoderma harzianum*, and *Trichoderma konigii*. In this study, the treatments are abbreviated as follows:

$T_1$ (control with no fertilizer);
$T_2$ (Ekosolfarm);
$T_3$ (Ekosolfarm + RhizoFill);
$T_4$ (Ekosolfarm + Subtima);
$T_5$ (Ekosolfarm + Fontera microzone);
$T_6$ (Ekosolfarm + Endo Roots Soluble—ERS);
$T_7$ (Ekosolfarm + Bontera).

A schedule was established for watering the plants with liquid vermicompost and other microbial fertilizers via drip irrigation. Precisely, the plants were watered with the liquid vermicompost and the microbial fertilizers at a rate of 1 mL L$^{-1}$ every 15 days via drip irrigation, beginning one week after planting [14].

### 2.2. Soil Analyses, Cultural Practices, and Measurements of Plant Parameters

Soil analysis was made prior to the trial. The application area had a loamy soil structure with a pH of 7.9 (Table 1).

The soil was covered with black polyethylene mulch. Except for the control ($T_1$) in the other six applications ($T_2$, $T_3$, $T_4$, $T_5$, $T_6$, $T_7$), 30 g of granular vermicompost Ekosalfarm was applied per plant by mixing it with the soil during the seedling planting. The AMF-containing $T_6$ treatment was used once at 0.2 g of ERS for each seedling by mixing it into the soil during the seedling planting. Fresh strawberry seedlings were planted in $30 \times 30$ cm using the triangle planting method in November 2019. The experiment was established in a randomized blocks experimental design with 4 replications and 20 plants in each replication (Figure 1). A low polyethene tunnel was established to ensure the

seedlings' survival during winter. The tunnels protected the plants from harsh weather conditions until March. The gradual opening of the tunnels ensured that the plants were gradually exposed to the external environment, avoiding shock to the plants' systems. The experiment was concluded at the end of May.

**Table 1.** Soil properties of the trial area.

| Soil Properties | Value |
|---|---|
| Texture | loamy |
| pH | 7.9 |
| Saltiness (%) | 0.05 |
| Lime (%) | 37.13 |
| Organic matter (%) | 0.51 |
| $P_2O_5$ (g m$^{-2}$) | 2.97 |
| $K_2O$ (g m$^{-2}$) | 37.77 |
| Ca (%) | 0.7033 |
| Mg (%) | 0.0358 |
| Na (%) | 0.0056 |
| Fe (mg kg$^{-1}$) | 1.99 |
| Cu (mg kg$^{-1}$) | 1.16 |
| Mn (mg kg$^{-1}$) | 0.94 |
| Zn (mg kg$^{-1}$) | 0.28 |

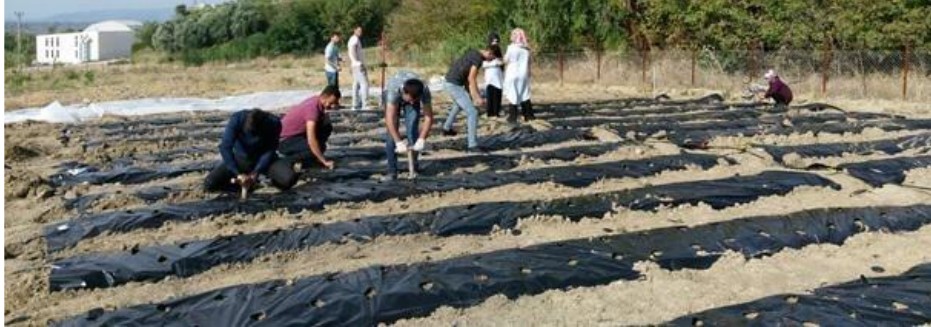

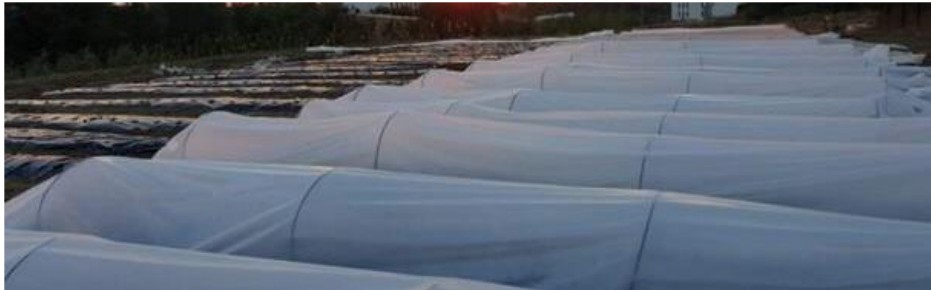

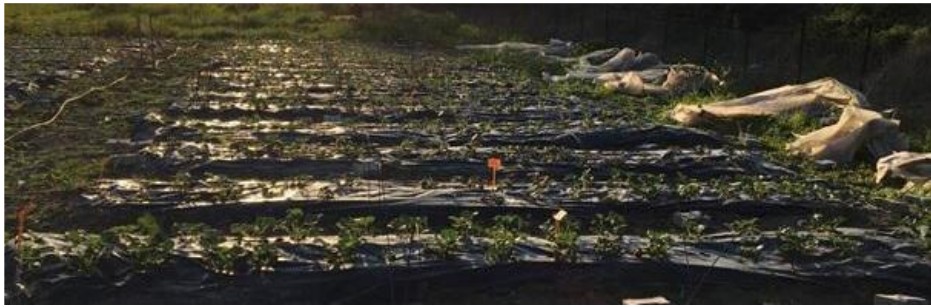

**Figure 1.** General images of seedling planting and low tunnels in organic strawberry cultivation experiment.

Plant vegetative growth parameters were measured at the end of the experiment using ten plants for each replication. The primary stem diameter, most extended root length and thickness, leaf area, and shoot and root dry matter per plant were determined. At the end of the experiment, the whole plant leaf area was measured by the Digimizer software version 5.3.5 in three randomly selected plants in each replication. In addition, the root length, root thickness, stem diameter, root, and stem dry matter content were determined in three randomly selected plants from each replication of the treatments. Their fresh weights were recorded initially to assess the root and stem dry matter. Then, they were dried in an oven at 65 °C until a constant weight was reached, and the dry matter ratios were calculated [14].

The strawberry harvest started in March and continued until the end of May when the weather warmed up. The fruit yield was recorded in the weekly strawberry harvests. In the middle of the harvest period in April, 20 fruits at the fully ripened stage were randomly taken from each replication for pomological analysis. The fruit weight, pH, total soluble solids (TSSs), and titrable acidity were measured according to Saygı [22]. The pH of the fruit was measured from 100 mL of strawberry juice using a digital pH meter (WTW pH 3110, Weilheim, Germany). The total soluble solids (TSSs) were measured with a temperature-compensated digital refractometer (Atago PR-101, Tokyo, Japan) at 25 °C and the results were expressed as % °Brix. The titratable acidity (TA) was measured via potentiometric titration against 0.1 N NaOH up to a pH of 8.1. All the determinations were made in triplicate and the results were expressed as g of citric acid equivalent per 100 g of fresh weight.

### 2.3. Determination of Leaf Minerals by Atomic Absorption Spectrophotometry

In April, leaf samples, 15 fully mature leaves of 10 plants per replicate, were collected for mineral nutrient analysis. The leaves were dried in a forced-air oven at 65 °C for 48 h and ground through a 40-mesh sieve for elemental analysis [14]. The samples were dry-ashed in a muffle furnace at 550 °C for six hours. After the samples became ash, 3.3% of HCl acid was added to the ashes, and the obtained solution was filtered using blue band filter papers. Potassium (K), calcium (Ca), magnesium (Mg), iron (Fe), manganese (Mn), zinc (Zn), and copper (Cu) concentrations were determined using an atomic absorption spectrophotometer [29].

### 2.4. Determination of Leaf Phosphorus

The dry-ashed, furnaced, and dissolved leaf samples (as mentioned above) were reacted with Barton's solution. The phosphorus (P) concentration was determined at a wavelength of 430 nm in the spectrophotometer [14,29].

### 2.5. Determination of Leaf Total Nitrogen

A 0.2 g ground sample was digested with 5 mL of concentrated $H_2SO_4$ at 380 °C using a selenium tablet in the combustion unit of a Kjeldahl apparatus for 1 h until the color turned pale. Then, distillation was performed with 28% NaOH according to a standard Kjeldahl protocol and then titration was performed with 0.01 N $H_2SO_4$. The total nitrogen (N) of the leaf was calculated with the amount of $H_2SO_4$ consumed in the titration [14,29].

### 2.6. Statistical Analyses

Data were subjected to the MSTAT-C software for the statistical analysis. The LSD (Least Significant Difference) test was used to determine the differences between the means at $p \leq 0.05$.

## 3. Results and Discussion

### 3.1. Plant Growth Parameters

The synergistic effect of vermicompost and microbial fertilizers on plant growth can be attributed to several key parameters. The application of vermicompost, PGPR, and AMF leads to elevated levels of essential plant growth regulators, such as cytokinins, auxins,

and humic acids. These bioactive compounds play a significant role in enhancing the plant height and leaf area, as well as the dry weights of shoots and roots [20]. The increased availability of nutrients facilitated by these organic amendments stimulates the activity of enzymes involved in chlorophyll synthesis and photosynthesis. Consequently, this biochemical activation contributes to the overall improvement in plant growth [20].

The effects of the applications on the root length, root thickness, root and stem dry matter ratio, and stem diameter were statistically significant (Table 2). The longest root was in $T_7$ (27.49 cm), followed by the $T_5$ (26.28 cm) and $T_3$ (25.73 cm) treatments, respectively. The root lengths of all the biofertilizer treatments were higher than the control ($T_1$). Ciylez and Esitken [30] reported a root length between 17.50 and 27.16 cm, and the biofertilizers generally increased vegetative plant growth.

**Table 2.** The effects of treatments on root length, thickness, root and stem dry matter, and stem diameter.

| Treatments | The Longest Root (cm) | The Thickest Root (mm) | Root Dry Matter (%) | Shoot Dry Matter (%) | Stem Diameter (mm) |
|---|---|---|---|---|---|
| $T_1$ | 22.06 c | 0.89 e | 33.29 c | 20.97 d | 11.71 c |
| $T_2$ | 24.84 b | 0.94 d | 40.56 b | 27.75 c | 13.04 b |
| $T_3$ | 25.73 ab | 1.04 c | 42.98 ab | 31.54 bc | 13.03 b |
| $T_4$ | 24.52 b | 1.06 c | 40.03 b | 32.12 b | 13.20 ab |
| $T_5$ | 26.28 ab | 1.12 b | 45.75 a | 35.62 b | 13.27 ab |
| $T_6$ | 25.08 b | 1.11 b | 43.30 ab | 35.79 b | 13.45 ab |
| $T_7$ | 27.49 a | 1.22 a | 45.81 a | 44.35 a | 14.06 a |
| $LSD_{0.05}$ | 1.88 | 0.03 | 3.98 | 4.36 | 0.94 |

The means, which differ significantly at the 5% level according to the LSD test, are shown with different letters. ($T_1$) no fertilizer (control); ($T_2$) Ekosolfarm; ($T_3$) Ekosolfarm + RhizoFill; ($T_4$) Ekosolfarm + Subtima; ($T_5$) Ekosolfarm + Fontera microzone; ($T_6$) Ekosolfarm + Endo Roots Soluble; ($T_7$) Ekosolfarm + Bontera.

When the root thickness data were examined, it was determined that the highest value, similar to the root length, was in the $T_7$ (1.22 mm) treatment, and the lowest was 0.89 mm in the control ($T_1$). The stem diameters of all the treatments were higher than the control. In addition, we observed that the plants with the thickest stems provided higher yields, as was the case in the $T_7$ application (14.06 mm). Similarly, Balci [31] reported a 13 mm stem diameter in an organically grown strawberry plant.

The highest amount of dry matter ratio in roots was revealed in the $T_7$ (45.81%) and $T_5$ (45.75%) treatments and the lowest in the control ($T_1$) (33.29%). In this study, the highest stem dry matter percentage was observed in $T_7$ (44.35%), followed by $T_6$ (35.79%) and $T_5$ (35.62%). The stem dry matter ratios of all treatments were higher than the control. Burgut et al. [32] stated that the highest dry matter ratios were 25% and 12.5% in roots and stems, respectively, in organically fertigated strawberry plants.

Shoot, root, and leaf growth characteristics were significantly affected by the applications and these effects are probably due to the combined effects of vermicompost and biofertilizers. Additionally, factors such as biofertilizer content, dosage, application time, and soil characteristics could be inter-related aspects.

The effect of the treatments on the leaf area was statistically significant. The maximum leaf area was measured in $T_7$ (667.03 cm$^2$ per plant$^{-1}$) followed by $T_5$ with 620.55 cm$^2$ per plant$^{-1}$ (Figure 2). Ipek et al. [33] reported that the highest leaf area value was 465.9 cm$^2$ using PGPR in conventional growing, while Khalil and Agah [34] stated that the maximum leaf area was 529.66 cm$^2$ by the application of 50% mineral fertilizer + *Trichoderma* and *Bacillus* strains. In our study, the increase in leaf area in the vermicompost with PGPR and AMF may have enabled the plant to efficiently uptake more water and nutrients from the soil. This, in turn, may have increased the number of fruits, weight, and yield due to better root, stem, and leaf development and higher photosynthesis. Seema et al. [35] noted that the increase in leaf area might be due to plant-growth-promoting rhizobacteria's ability to produce plant growth hormones, better root development, better displacement, and an efficient use of water and nutrients.

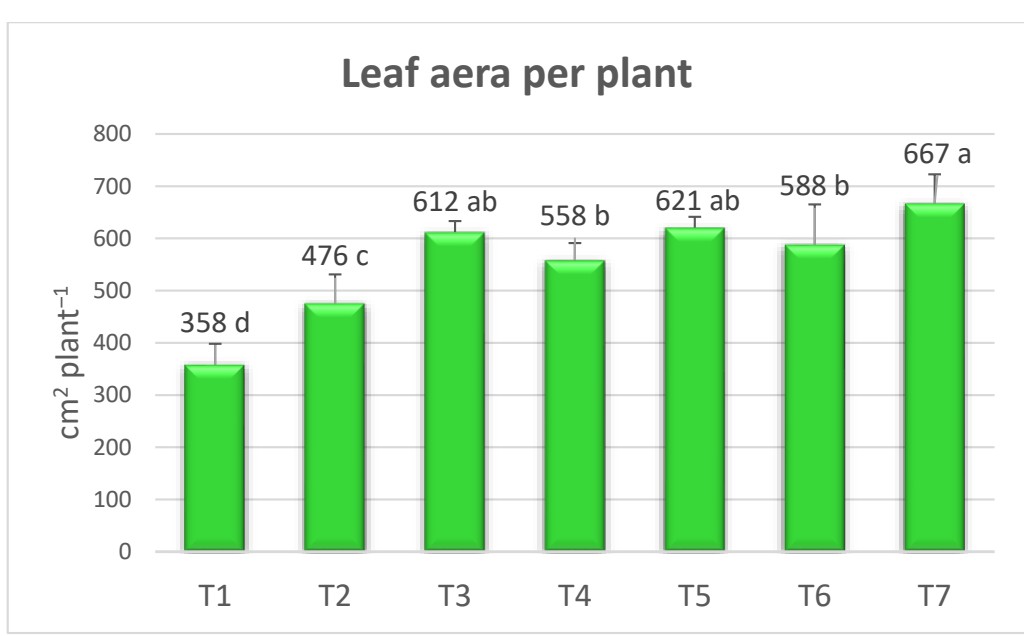

**Figure 2.** Effects of the treatment on leaf area of organically grown strawberry plants. The means, which differ significantly at the 5% level according to the LSD test, are shown with different letters. ($T_1$) no fertilizer (control); ($T_2$) Ekosolfarm; ($T_3$) Ekosolfarm + RhizoFill; ($T_4$) Ekosolfarm + Subtima; ($T_5$) Ekosolfarm + Fontera microzone; ($T_6$) Ekosolfarm + Endo Roots Soluble (Mycorrhiza); ($T_7$) Ekosolfarm + Bontera.

### 3.2. Total Yield and Average Fruit Weight per Plant

The differences between the treatments in terms of total yield per plant were statistically significant. As seen in Figure 3, the highest total yield was promoted in the $T_7$ treatment with 216.75 g per $plant^{-1}$ and the lowest yield was obtained in the control ($T_1$) treatment with 108.36 g per $plant^{-1}$. The yield increase in $T_7$ (Ekosolfarm + Bontera) was 100.03% compared to no fertilizer ($T_1$) and 29.09% compared to vermicompost alone ($T_2$). Compared to vermicompost alone ($T_2$), the yield was higher in other treatments of the combined use of the vermicompost with PGPR and AMF ($T_3$, $T_4$, $T_5$, $T_6$, $T_7$). Therefore, the growth in yield may be due to the fact that vermicompost, PGPR, and AMF facilitate the plant's nutrient uptake. Imriz et al. [25] and Seema et al. [35] have reported that the application of biofertilizers containing auxins and cytokinins can potentially influence various physiological processes of plants, such as cell division, flower and root development, and leaf growth. Moreover, biofertilizers are believed to enhance the uptake and utilization of water and nutrients in plants, ultimately leading to increased fruit and flower production and higher yields. Similarly, previous studies have reported that the application of vermicompost and Azotobacter in the "Sweet Charlie" strawberry cultivar resulted in the highest yield of 124.46 g per $plant^{-1}$ [6], while the combined application of vermicompost, poultry manure, and Azotobacter induced a yield of 144.77 g per $plant^{-1}$ [8]. The "Chandler" strawberry cultivar produced the highest yield of 185.08 g per $plant^{-1}$ in the treatment where organic manure and plant-growth-promoting rhizobacteria (PGPR), such as *Azotobacter chroococcum*, and *Pseudomonas fluorescens*, were applied in combination [9]. Moreover, we observed findings similar to those of the researchers mentioned above.

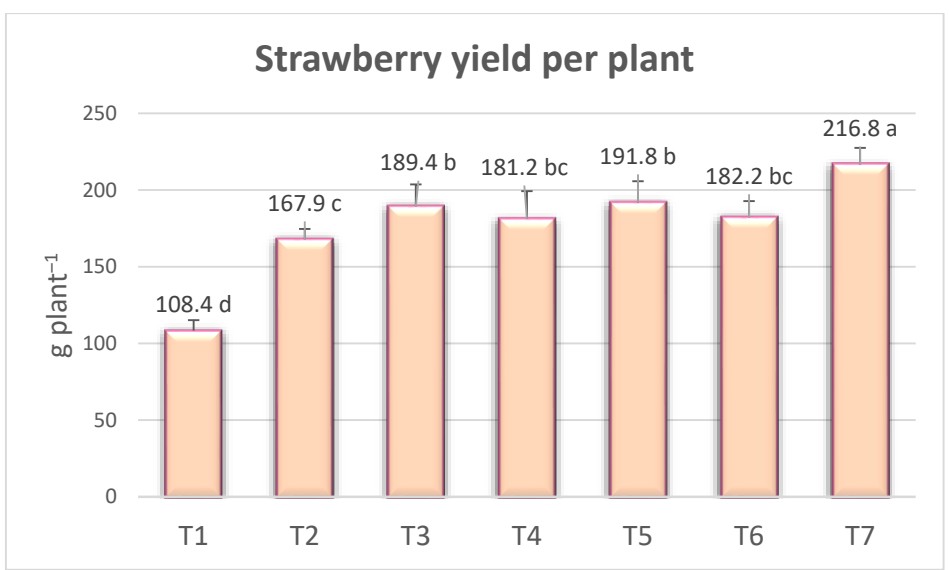

**Figure 3.** The effect of treatments on strawberry yield per plant. The means, which differ significantly at the 5% level according to the LSD test, are shown with different letters. ($T_1$) no fertilizer (control); ($T_2$) Ekosolfarm; ($T_3$) Ekosolfarm + RhizoFill; ($T_4$) Ekosolfarm + Subtima; ($T_5$) Ekosolfarm + Fontera microzone; ($T_6$) Ekosolfarm + Endo Roots Soluble (Mycorrhiza); ($T_7$) Ekosolfarm + Bontera.

The differences in the fruit weight ratios of the treatments were statistically significant (Figure 4). All treatments were more effective on fruit weights than the control ($T_1$). Enormous fruits were observed in the $T_7$ treatment (18.11 g) and minor fruits in the control ($T_1$) (15.51 g). In the $T_7$ treatment, the fruit weight increased by 16.76% compared to no fertilizer ($T_1$) and 7.41% compared to vermicompost alone ($T_2$). Seema et al. [35] reported that the maximum fruit weight in the PGPR biofertilizer treatments in the Chandler strawberry variety was 14.62 g. Negi et al. [9] reported that the highest fruit weight in organic manure with PGPR treatments in the same variety was 13.30 g. Although fruit size is a variety-specific feature, environmental conditions, applied fertilizers, and treatment methods can be effective.

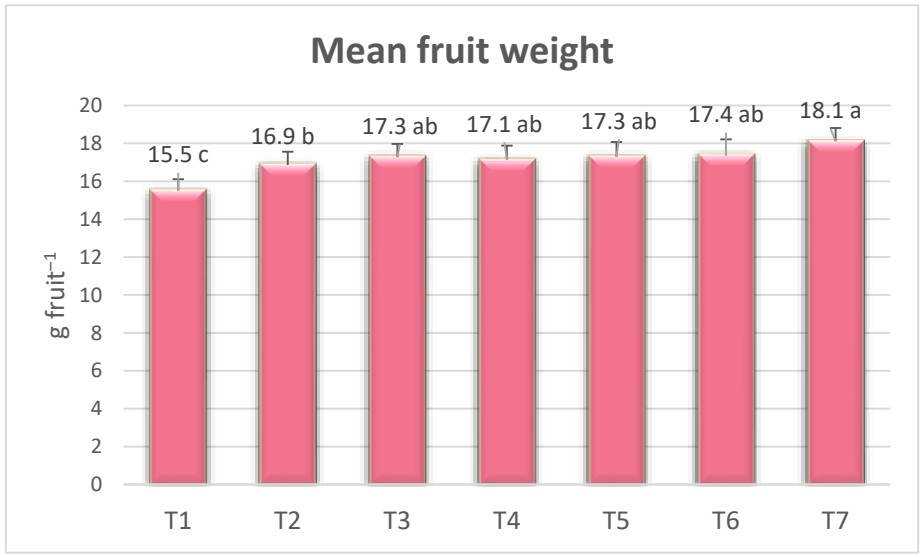

**Figure 4.** The effect of the treatments on fruit weight. The means, which differ significantly at the 5% level according to the LSD test, are shown with different letters. ($T_1$) no fertilizer (control); ($T_2$) Ekosolfarm; ($T_3$) Ekosolfarm + RhizoFill; ($T_4$) Ekosolfarm + Subtima; ($T_5$) Ekosolfarm + Fontera microzone; ($T_6$) Ekosolfarm + Endo Roots Soluble (Mycorrhiza); ($T_7$) Ekosolfarm + Bontera.

Additionally, researchers have suggested that the increase in fruit size may be attributed to the beneficial microorganisms present in vermicompost. The enhancement of the nutrient uptake may have promoted the production of phytohormones, particularly gibberellins, leading to increased nutrient and water absorption by the plants [8,13,15,36].

Fruit growth can be characterized by two distinct stages: cell division and cell enlargement. The initial phase involves cell division, followed by a subsequent process of cell volume increase. The influence of gibberellin on fruit enlargement primarily occurs during the rapid growth period in the first phase of fruit development. At this stage, the presence of gibberellin facilitates the transition from cell division to cell enlargement and also facilitates the mobilization, transportation, and accumulation of nutrients within the fruit [8,24]. Additionally, gibberellin induces an increase in auxin content, which enhances nutrient absorption by the fruit and promotes growth in overall fruit size [8,24]. Moreover, we observed findings similar to those of the researchers mentioned above. The extra beneficial microorganisms and organic matter in the vermicompost content increased the yields and fruit sizes of the strawberry plants.

The strawberry yield results obtained were dependent on the composition, application methods, and dosages of the vermicompost and biofertilizers used, as well as the soil and climatic characteristics of the experimental site.

### 3.3. Fruit pH, Total Soluble Solids (TSSs), Titrable Acid, and the Ratio of TSSs/Acid

Among all the treatments, $T_7$ provided the highest pH (3.95) in strawberry fruits. The lowest fruit juice pH was observed in the control treatment ($T_1$) at 3.83, followed by the $T_2$ treatment (3.87). The differences in pH results were statistically significant (Table 3). Jain et al. [37] stated that the pH of strawberries varied between 2.63 and 3.90, while Sener and Duran [38] reported that the pH values were between 3.59 and 3.81. Strawberry pH values may also vary according to the region's climate, the plant's genetic material, and fertilizer [22].

**Table 3.** The effect of treatment on pH, TSSs, acidity, and TSSs/acid ratio of the strawberry fruit.

| Treatments | pH | TSSs (%) | TA (%) | TSSs/TA |
|---|---|---|---|---|
| $T_1$ | 3.83 e | 9.48 e | 0.67 a | 14.27 e |
| $T_2$ | 3.87 d | 9.82 d | 0.61 b | 16.17 d |
| $T_3$ | 3.92 b | 10.62 a | 0.62 b | 17.24 bc |
| $T_4$ | 3.90 bc | 10.07 cd | 0.61 b | 16.54 cd |
| $T_5$ | 3.91 b | 10.23 bc | 0.57 c | 18.07 ab |
| $T_6$ | 3.89 cd | 10.06 cd | 0.60 b | 16.86 cd |
| $T_7$ | 3.95 a | 10.42 ab | 0.56 c | 18.74 a |
| $LSD_{0.05}$ | 0.02 | 0.29 | 0.03 | 0.91 |

The means, which differ significantly at the 5% level according to the LSD test, are shown with different letters. ($T_1$) no fertilizer (control); ($T_2$) Ekosolfarm; ($T_3$) Ekosolfarm + RhizoFill; ($T_4$) Ekosolfarm + Subtima; ($T_5$) Ekosolfarm + Fontera microzone; ($T_6$) Ekosolfarm + Endo Roots Soluble (Mycorrhiza); ($T_7$) Ekosolfarm + Bontera; TSSs: total soluble solids; TA: titratable acidity.

The effects of vermicompost alone or combined with the PGPR and AMF on TSSs were statistically significant (Table 3). $T_3$ showed the highest TSSs, 10.62%, followed by $T_7$ at 10.42%, while the lowest TSSs were measured in the control treatment ($T_1$), 9.48%. Similarly, Negi et al. [9] stated that the highest TSSs value in the "Chandler" strawberry cultivar was observed in organic manure with a PGPR treatment, 10.47%, and the lowest in the control treatment, 7.60%. The researchers additionally stated that the observed rise in TSSs levels could be attributed to the fast metabolic conversion of the manure, vermicompost, and biofertilizer combination into soluble starch and pectin compounds. Furthermore, they suggested that the transition of sugar from leaves to developing fruits might result from converting polysaccharides into simpler sugars. Table 3 shows that the treatments significantly affected the titrable acidity of the strawberries. For instance, the lowest acidity was obtained in $T_7$, 0.56%, and $T_5$, 0.57%, and the highest in the control ($T_1$), 0.67%. Singh et al. [39] observed that the acidity values were between 0.35 and 0.65%, while

Anuradha et al. [40] reported that the values varied between 0.73 and 0.84%. TSSs/acidity ratios determined strawberries' taste and were statistically significant in our study.

Strawberry consumers prefer fruits characterized by a high sugar content, specifically total soluble solids (TSSs), and low acidity levels. The augmentation of TSSs in strawberry fruits through the activity of plant-growth-promoting Rhizobacteria (PGPR) and arbuscular mycorrhizal fungi (AMF) during cultivation is likely attributed to the provision of nutrients by these microorganisms. This nutrient supply enhances the vigor of strawberry plants, leading to an increase in leaf area (Figure 2) and a subsequent boost in the synthesis of photo-assimilates, resulting in an elevated rate of photosynthesis [41]. In addition, the high mobility of photosynthetic products from leaves to developing fruits contributes to the rise in TSSs levels within the fruits [41]. However, the lower TSSs values observed in the control group ($T_1$) and the vermicompost alone treatment ($T_2$) can be attributed to the limited availability of metabolites primarily allocated for vegetative development, leaving only a tiny portion for fruit development [41]. This limitation in photosynthetic capacity and nutrient uptake may account for the reduced TSSs in these treatments.

The titratable acidity reached its maximum level in the control group, which received no fertilizer. The application of vermicompost alone or a combined inoculation of PGPR and AMF reduced titratable acidity. This decline in acidity can be attributed to the presence of nitrogen, potassium, calcium, magnesium, and micronutrients, as changes in their availability directly influence fruit quality, particularly titratable acidity, which tends to increase when the availability of these nutrients decreases (Tables 3 and 4) [41]. In our experiment, a significant increase in nutrient availability (Tables 4 and 5) was observed in plots treated with vermicompost alone, especially in those treated with combined PGPR and AMF. Fruit pH is inversely related to acidity; therefore, higher acidity corresponds to lower pH values.

**Table 4.** The effect of the treatments on the macroelement concentrations of strawberry leaves (%).

| Treatments | N | P | K | Ca | Mg |
|---|---|---|---|---|---|
| $T_1$ | 2.63 e | 0.14 d | 0.91 c | 1.61 c | 0.19 c |
| $T_2$ | 3.00 d | 0.33 c | 1.33 b | 1.98 b | 0.25 b |
| $T_3$ | 3.24 cd | 0.68 a | 1.37 ab | 2.07 b | 0.25 b |
| $T_4$ | 3.19 cd | 0.34 c | 1.34 b | 2.03 b | 0.25 b |
| $T_5$ | 3.68 b | 0.43 b | 1.38 ab | 2.37 a | 0.26 b |
| $T_6$ | 3.30 c | 0.48 b | 1.41 ab | 2.04 b | 0.25 b |
| $T_7$ | 3.97 a | 0.70 a | 1.46 a | 2.47 a | 0.30 a |
| LSD$_{0.05}$ | 0.25 | 0.07 | 0.11 | 0.23 | 0.02 |

The means, which differ significantly at the 5% level according to the LSD test, are shown with different letters. ($T_1$) no fertilizer (control); ($T_2$) Ekosolfarm; ($T_3$) Ekosolfarm + RhizoFill; ($T_4$) Ekosolfarm + Subtima; ($T_5$) Ekosolfarm + Fontera microzone; ($T_6$) Ekosolfarm + Endo Roots Soluble (Mycorrhiza); ($T_7$) Ekosolfarm + Bontera.

**Table 5.** The effect of treatments on microelement concentrations of strawberry leaves (ppm).

| Treatments | Fe | Zn | Mn | Cu |
|---|---|---|---|---|
| $T_1$ | 52.00 e | 27.00 d | 34.50 c | 3.00 d |
| $T_2$ | 73.30 d | 32.75 c | 52.50 b | 4.00 c |
| $T_3$ | 110.00 a | 51.56 a | 58.00 ab | 5.67 a |
| $T_4$ | 85.00 bc | 42.67 b | 61.75 a | 5.00 b |
| $T_5$ | 92.67 b | 42.33 b | 57.25 ab | 5.00 b |
| $T_6$ | 84.00 c | 40.56 b | 58.25 ab | 4.00 c |
| $T_7$ | 84.11 c | 50.67 a | 63.00 a | 4.00 c |
| LSD$_{0.05}$ | 8.28 | 5.11 | 8.86 | 0.67 |

The means, which differ significantly at the 5% level according to the LSD test, are shown with different letters. ($T_1$) no fertilizer (control); ($T_2$) Ekosolfarm; ($T_3$) Ekosolfarm + RhizoFill; ($T_4$) Ekosolfarm + Subtima; ($T_5$) Ekosolfarm + Fontera microzone; ($T_6$) Ekosolfarm + Endo Roots Soluble (Mycorrhiza); ($T_7$) Ekosolfarm + Bontera.

The TSSs/acidity ranged between 14.27 and 18.74, with the highest TSSs/acidity ratio in the $T_7$ treatment (Ekosolfarm + Bontera), 18.74 (Figure 3). These confirm the values from

the literature. Kumar et al. [6] reported that the TSSs/acidity ratio in the "Sweet Charlie" cultivar varied between 12.84 and 17.05 in applications of vermicompost and biofertilizer treatments. Similarly, Kumar et al. [36] found that the TSSs/acidity ratio in the "Chandler" cultivar ranged from 11.3 to 14.0.

*3.4. Plant Nutrient Analyses in Leaves*

One crucial finding of this study demonstrates that relying solely on vermicompost-based fertilization is inferior, and the use of complementary top dressing with biofertilization seems to be necessary for superior plant nutrition, especially in $T_7$ conditions. The concentrations of macro- and micronutrients in strawberry leaves could be influenced by the combination of vermicompost and the biofertilizers, their application ratios and timing, and the soil properties employed in the experiment. The effects of the vermicompost alone or in combination with PGPR and AMF on the leaves' total nitrogen, phosphorus, potassium, calcium, and magnesium concentrations were statistically significant (Table 4). The leaf nitrogen concentration varied between 2.63% and 3.97%, and the highest nitrogen among all the treatments was revealed in $T_7$ (3.97%). Nitrogen concentration increases in $T_2$–$T_7$ applications ranged from 14% to 51% compared to the control ($T_1$). Jones et al. [42] indicated that the nitrogen adequacy level in the strawberry leaf was between 2.5 and 4.00%. Therefore, the nitrogen concentration of all treatments was within the adequacy limits. The increase in nitrogen may be due to the fixation of air nitrogen by the PGPR, as well as their facilitation and uptake of the nutrient from the soil [25].

Meanwhile, Joshi et al. [19] observed the highest nitrogen content in the leaves in a vermicompost + *Azotobacter* treatment (3.31%). He reported that the reasons for the increase were biological nitrogen fixation, enzyme complex production, and the dissolved nutrient form, which was readily available to be used. Tomic et al. [43] reported that the leaf nitrogen varied between 1.94 and 2.03%, while Negi et al. [9] noted that the leaf nitrogen content of the Chandler strawberry cultivar was between 0.81 and 2.67% in a manure and biofertilizer treatment.

In this study, the leaf phosphorus concentration was between 0.14 and 0.70%, and the highest P was in $T_7$ (0.70%) and $T_3$ (0.68%) treatments, while the lowest P was in the control ($T_1$) with 0.14%. Phosphorus concentration increases in $T_2$–$T_7$ treatments ranged from 135% to 400% compared to the control ($T_1$). Jones et al. [42] stated that the phosphorus sufficiency level in leaves was between 0.25 and 1.00%. Although the phosphorus content in the leaves was within the appropriate limits in vermicompost alone or in combination with PGPR and AMF, it was insufficient in the control treatment ($T_1$). Accordingly, the increase in the phosphorus in the leaves may be due to the phosphorus solubility feature of the biofertilizers. Similarly, Imriz et al. [25] stated that organic acids are essential in the mineralization of organic phosphorus in the soil as it is equally critical to make it readily available for use by the plant. They also noted that *Bacillus subtilis*, *Bacillus polymyxa*, *B megatarium*, *Pseudomonas striata*, *P. rathonia*, and *Rhizobium leguminosarum* are among the bacteria that can dissolve phosphorus by producing organic acids. Ipek et al. [33] reported that leaf phosphorus values were between 0.27 and 0.41% with PGPR application. Basu et al. [27] indicated that an essential feature of biofertilizers is the solubilization and mineralization of phosphorus, which converts it into a usable form for plants.

We also observed that the leaf potassium was between 0.91–1.46% in our trial. The potassium levels of $T_7$ (1.46%) were higher than in other treatments. Potassium concentration increases in $T_2$–$T_7$ applications ranged from 46% to 60% compared to the control ($T_1$). Jones et al. [42] pointed out that the potassium sufficiency level in the leaves is between 1.30 and 3.00%. While the potassium was sufficient in vermicompost alone and combined with PGPR and AMF treatments, it was insufficient in the control ($T_1$).

The leaf calcium levels observed in the treatments were between 1.61 and 2.47%, while the highest calcium concentrations were in $T_7$ and $T_5$ (2.47% and 2.37%, respectively) and the lowest was 1.61% in the control ($T_1$). Calcium concentration increases in $T_2$–$T_7$ applications ranged from 23% to 53% compared to the control ($T_1$). Beer and Singh [44]

reported the highest calcium in vermicompost with a phosphate solubilizing biofertilizer treatment (2.50%) and the lowest in a control treatment (1.95%). Jones et al. [42] that the calcium sufficiency level in leaves was between 1.00 and 2.50%. Likewise, the calcium contents of the treatments of the current study are within the sufficiency limits.

The total magnesium concentrations in the leaves ranged from 0.19 to 0.30%, and the highest magnesium was 0.30% in the $T_7$ treatment. Magnesium concentration increases in $T_2$–$T_7$ applications ranged from 32% to 58% compared to the control ($T_1$). Jones et al. [42] stated that the magnesium sufficiency level in leaves was between 0.25 and 1.00%. In this study, while the amount of magnesium was sufficient in vermicompost alone and combined with PGPR and AMF treatments, it was insufficient in the control ($T_1$).

Table 5 shows that the iron, zinc, manganese, and copper concentrations in leaves are statistically significant. The highest iron concentration was detected in the $T_3$ treatment, 110 ppm, and the lowest in the control ($T_1$) at 52 ppm. Iron concentration increases in $T_2$–$T_7$ treatments ranged from 41% to 62% compared to the control ($T_1$). Jones et al. [42] reported that leaf iron in strawberries was sufficient between 50 and 200 ppm, and Ipek et al. [33] stated that it was between 46.1 and 85.7 ppm in their study with PGPR.

The zinc concentration in the leaves was between 27 and 51.56 ppm. The highest zinc concentrations were observed in $T_3$ (51.56 ppm) and $T_7$ (50.67 ppm), while the lowest zinc was in the control treatment ($T_1$) (27 ppm). Increases in zinc concentration in $T_2$–$T_7$ applications ranged from 21% to 88% compared to the control ($T_1$). Jones et al. [42] pointed out that the zinc sufficiency level in leaves was between 20 ppm and 200 ppm.

The highest manganese levels were in the $T_7$ and $T_4$ treatments (63.00–61.75 ppm, respectively), and the lowest Mn was in the control ($T_1$), 34.50 ppm. Manganese concentration increases in $T_2$–$T_7$ treatments ranged from 34% to 83% compared to the control ($T_1$). Jones et al. [42] stated that the manganese sufficiency level in leaves was between 50 and 200 ppm. The manganese concentration was observed to be sufficient in vermicompost alone and combined with PGPR and AMF, yet it was insufficient in $T_1$ (control).

The copper concentration ranged between 3.00 and 5.67 ppm, with the highest value in $T_3$ (5.67 ppm). Copper concentration increases in $T_2$–$T_7$ treatments ranged from 33% to 89% compared to the control ($T_1$). Jones et al. [42] reported that the copper sufficiency level in the leaf is between 6 and 50 ppm. Accordingly, the concentrations of copper observed in the current study were insufficient and should be pointed out for further studies.

The contributions of PC1 and PC2 to variability were 85.3% and 8.3%, respectively (Figure 5). The principal component analysis (PCA) based on an analysis of twenty-one variables has scattered the $T_1$–$T_7$ treatments. The control ($T_1$) and the vermicompost alone ($T_2$) treatments disintegrated and dissociated from the complementary biofertilizer PGPR and AMF applications. The complementary biofertilizer PGPR and AMF applications have been grouped and affiliated in the scatterplot.

Heat maps provide a visual tool for comprehending numerical values by presenting them in a graphical format. This approach enables the simultaneous depiction of multiple data points and their inter-relationships within a single image, employing a color-coded system [15]. In Figure 6, the collective impact of vermicompost-based fertilization and microbial complementary nutrition on plant growth, yield, fruit quality, and plant nutrition parameters can be observed effectively through heat maps. Furthermore, heat maps can be utilized literally, such as illustrating "hot and cold" areas (e.g., from red to blue) on a map. Among the treatments examined, "$T_7$: Ekosolfarm + Bontera" exhibited the highest prevalence of warm red tones, signifying favorable outcomes. Conversely, applying "$T_1$: No-fertilizer control" resulted in predominantly cold, dark-blue hues, indicating fewer desirable results. The relationships between the remaining treatments ($T_2$, $T_3$, $T_4$, $T_5$, and $T_6$) and the examined parameters are demonstrated in Figure 6, based on the intensity of red and blue shades. Consequently, the effectiveness of vermicompost-based fertilization and microbial complementary nutrition is visually categorized and interpreted.

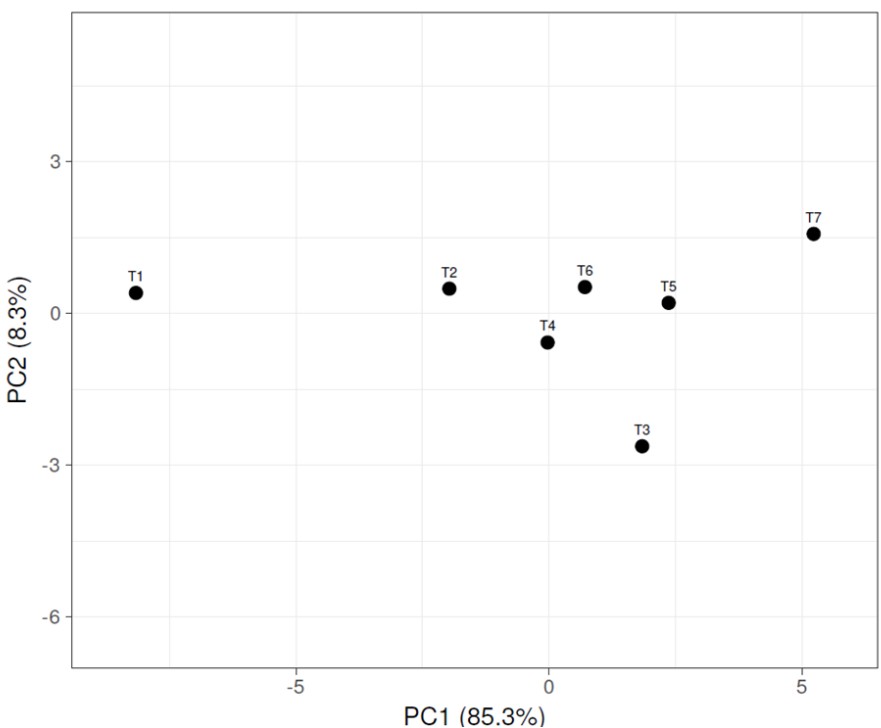

**Figure 5.** The principal component analysis of the vermicompost-based fertilization and microbial complementary nutrition. (T$_1$) no fertilizer (control); (T$_2$) Ekosolfarm; (T$_3$) Ekosolfarm + RhizoFill; (T$_4$) Ekosolfarm + Subtima; (T$_5$) Ekosolfarm + Fontera microzone; (T$_6$) Ekosolfarm + Endo Roots Soluble (Mycorrhiza); (T$_7$) Ekosolfarm + Bontera.

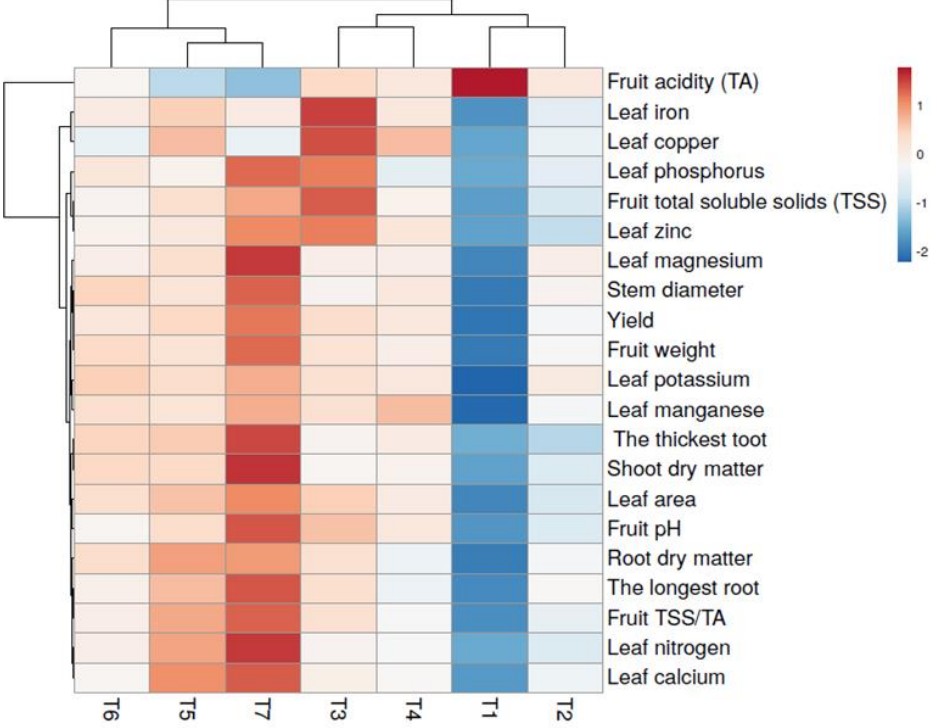

**Figure 6.** Heat map of the vermicompost-based fertilization and microbial complementary nutrition. (T$_1$) no fertilizer (control); (T$_2$) Ekosolfarm; (T$_3$) Ekosolfarm + RhizoFill; (T$_4$) Ekosolfarm + Subtima; (T$_5$) Ekosolfarm + Fontera microzone; (T$_6$) Ekosolfarm + Endo Roots Soluble (Mycorrhiza); (T$_7$) Ekosolfarm + Bontera.

## 4. Conclusions

In conclusion, this study demonstrates that combining vermicompost-based fertilization and complementary microbial nutrition is a practical approach to improving organic strawberries' growth, yield, plant nutrition, and fruit quality. Treatment $T_7$ (Ekosolfarm and Bontera) showed the most promising results among the biofertilizer combinations tested. These findings have significant implications for sustainable agriculture and organic farming. They offer a practical and sustainable solution to meet the high demand for organic strawberries while increasing producers' income. These results are promising for enhancing organic strawberry production. The outcomes of this study depend on several factors, including the species of PGPR and AMF present in the microbial fertilizers, the concentration of viable microorganisms, and the application rates and timing of the biofertilizers, as well as the physical and chemical properties of the experimental soil in which the strawberry plants were organically grown.

**Author Contributions:** All the authors contributed to this research. N.K. and H.Y.D. designed the experiment. Conceptualization; data curation; formal analysis; investigation; resources; funding acquisition, N.K. and H.Y.D. Supervision and writing—review and editing, H.Y.D., N.K. and N.S.G. All authors have read and agreed to the published version of the manuscript.

**Funding:** This research received no external funding.

**Data Availability Statement:** The data presented in this study are available in the article.

**Conflicts of Interest:** The authors declare no conflict of interest.

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
