# Peer review of "A Novel Approach for Organic Strawberry Cultivation: Vermicompost-Based Fertilization and Microbial Complementary Nutrition"

_horticulturae, doi:10.3390/horticulturae9060642_

Round 1

Reviewer 1 Report

The study presented a detailed analysis of the impacts of vermicompost-based fertilization, which was complemented with various microbial fertilizers, including different plant growth promotion rhizobacteria and arbuscular mycorrhizal fungi, on the growth performance and nutrient contents of strawberry. The efficacy of the different fertilizers on strawberry was evaluated based on a set of parameters. These findings are of great practical importance for sustainable organic strawberry production. The experiment was well-established and conducted, and the manuscript was well-written. My only suggestion would be to rank the efficacy of T1-T7 using statistical methods, such as membership function analysis, which would increase the logicality and readability of the manuscript. Other minor revisions are listed below.

Line 104. Delete the bracket at the end of the sentence.

Line 256. Please revise the sentence.

Lines 272-273. What does the “fruit size” mean in the text? The average weight of a fruit?

Lines 301-306. What’s the biological meanings for the shifts in pH, acidity, and TSS of strawberry fruit juice, except for its potential impact on the taste? This is important to evaluate the impacts of different fertilizers on fruit quality. The authors should explain the meaning of these changes in the text as a conclusion.

Line 343. Please add bracket in the text.

Author Response

RESPONSES TO REVIEWER 1

Dear reviewer,

We are pleased with your positive comments about our study findings being of great practical importance for sustainable organic strawberry production. Thank you very much. The changes are highlighted in the manuscript in red colour. Please find our answer to your question below in red colour.

Comments and Suggestions for Authors:

The study presented a detailed analysis of the impacts of vermicompost-based fertilization, which was complemented with various microbial fertilizers, including different plant growth promotion rhizobacteria and arbuscular mycorrhizal fungi, on the growth performance and nutrient contents of strawberry. The efficacy of the different fertilizers on strawberry was evaluated based on a set of parameters. These findings are of great practical importance for sustainable organic strawberry production. The experiment was well-established and conducted, and the manuscript was well-written. My only suggestion would be to rank the efficacy of T1-T7 using statistical methods, such as membership function analysis, which would increase the logicality and readability of the manuscript. Other minor revisions are listed below.

REPLY: BELOW, OUR ANSWERS WITH THE NECESSARY EXPLANATIONS

Our experimental design might not be technically suitable for “the membership function analysis”. Also, because the membership function analysis includes advanced mathematical and some artificial intelligence models, it would not be possible to implement it in the limited manuscript-revision process. However, we used the graphical model of the Heat Map and the Principal Component Analysis (PCA) to increase the logicality and readability of the manuscript. Lines 475-480 and Figure 5, Lines 490-504 and Figure 6 have been added to the manuscript with red colour. We hope these changes fulfil your requirements.

Lines 475-480: The contribution of PC1 and PC2 to variability was 85.3% and 8.3%, respectively (Figure 5). The principal component analysis (PCA) based on an analysis of twenty-one variables has scattered the T1-T7 treatments. The control (T1) and the vermicompost alone (T2) treatments disintegrated and dissociated from the complementary biofertilizers PGPR and AMF applications. The complementary biofertilizers PGPR and AMF applications have been grouped and affiliated in the scatterplot.

Line 483: Figure 5. The principal component analysis of the vermicompost-based fertilization and microbial complementary nutrition

(T1) No fertilizer (control); (T2) Ekosolfarm; (T3) Ekosolfarm+RhizoFill; (T4) Ekosolfarm+Subtima; (T5)Ekosolfarm+Fontera microzone; (T6) Ekosolfarm+Endo Roots Soluble (Mycorrhiza); (T7) Ekosolfarm+Bontera.

Line 506 Figure 6. Heat map of the vermicompost-based fertilization and microbial complementary nutrition

(T1) No fertilizer (control); (T2) Ekosolfarm; (T3) Ekosolfarm+RhizoFill; (T4) Ekosolfarm+Subtima; (T5) Ekosolfarm+Fontera microzone; (T6) Ekosolfarm+Endo Roots Soluble (Mycorrhiza); (T7) Ekosolfarm+Bontera.

OTHER RESPONSES TO MINOR REVISIONS ARE LISTED BELOW

  1. Line 104. Delete the bracket at the end of the sentence.

Reply L104: The bracket was deleted.

  1. Line 256. Please revise the sentence.

Reply L273: After the first sentence in this line, a point (.) was put, and the sentence was corrected.

  1. Lines 272-273. What does the “fruit size” mean in the text? The average weight of a fruit?

Reply L289: "Fruit size" is corrected to "fruit weight". We meant "fruit weight".

  1. Lines 301-306. What’s the biological meanings for the shifts in pH, acidity, and TSS of strawberry fruit juice, except for its potential impact on the taste? This is important to evaluate the impacts of different fertilizers on fruit quality. The authors should explain the meaning of these changes in the text as a conclusion.

Reply L348-371: The following paragraphs have been added to the manuscript:

Strawberry consumers like fruits with high sugar, TSS, and low acid content. Increased TSS in strawberry fruits by the PGPR and AMF might be due to the offer of nutrients by the microorganisms during cultivation, which increased the strength of strawberry plants and increased leaf area (Figure 2) with higher synthesis of photo-assimilates due to enhanced rate of photosynthesis [44]. The increased mobility rate of photosynthetic products from leaves to developing fruits might be attributed to the increasing TSS of the fruit [44].  However, lower TSS under control (T1) and the vermicompost alone (T2) might be because most of the metabolites were only enough for vegetative development. In contrast, a little amount was left for the fruit [44]. Because the photosynthetic capacity and nutrition of the plant might be limited.

The titratable acidity reached its maximum level in the control group, which received no fertilizer. The application of vermicompost alone or combined inoculation of PGPR and AMF reduced titratable acidity. This decline in acidity can be attributed to the presence of nitrogen, potassium, calcium, magnesium, and micronutrients, as changes in their availability directly influence fruit quality, particularly titratable acidity, which tends to increase when the availability of these nutrients decreases (Tables 3 and 4) [44]. In our experiment, a significant increase in nutrient availability (Tables 4 and 5) was observed in plots treated with vermicompost alone, especially in those treated with combined PGPR and AMF. Fruit pH is inversely related to acidity; therefore, higher acidity corresponds to lower pH values.

  1. Line 343. Please add bracket in the text.

Reply L397: We have added the bracket in the text.

Reviewer 2 Report

The study investigated the effects of vermicompost fertilization with the complementary  microbial nutrition on growth, yield and fruit quality of strawberry. The topic of the article is relevant.  The results are of great practical importance. However, there are comments on the text (in Article review).

Author Response

RESPONSES TO REVIWER 2

Dear reviewer,

Thank you very much for your valuable comments and suggestions. We accepted most of your suggestions. The changes are highlighted in the manuscript in red colour. Please find our answer to your question below in red colour.

  1. The originality of the research is not clear. Based on the literature references, it can be concluded that the originality of article only in the use of different strawberry variety.

Reply Lines 82-91 was added: The originality of this study is organic plant nutrition by "vermicompost-based fertilization and microbial complementary (top-dress) nutrition". In organic strawberry cultivation, using mineral chemical synthetic fertilizers is forbidden. For this reason, it is necessary to enrich the plant nutrition. PGPR and AMF bio-fertilizers were selected from commercial preparations for easy access by strawberry farmers. There are previous studies using vermicompost, PGPR and AMF. However, these studies are not organic farming. We tried to combine vermicompost and microbial fertilizers to find the most suitable combination for organic strawberry nutrition.

  1. In “3.1 Plant growth parameter” are not discussed the result of the experiments.

Reply Lines 213-220, the discussion was added:

The improvement in plant growth in the combination of vermicompost and microbial fertilizers could be refered to some parameters. The vermicompost, PGPR and AMF provide high levels of cytokinins, auxins, and humic acids, which could be an significant cause for enhancements in plant height, leaf area and dry weights of shoots and roots [20]. Increased content of nutrients could activate enzymes involved in chlorophyll synthesis and photosynthesis which in turn improved increases in plant growth [20].

  1. Р. 280-282. As a known the physiological effect of gibberellins is to stimulation of growth by increasing cell elongation and cell division of meristematic tissues. How can an increase of gibberellins synthesis contribute to an increase of fruit size?

Reply Line 300-307 the discussion was added:

Fruit growth can be divided into two stages: cell division and cell enlargement The early phase of fruit is cell division, and then it continues the process of cell volume increase.  The impact of gibberellin on fruit enlargement is mainly in the rapid growth period of the first phase of the fruit. During this period, the increment of gibberellin promotes the change of cell division to cell enlargement, and also mobilizes the transportation and accumulation of nutrients to the fruit [8, 24]. The gibberellin also induces an increase in the content of auxin, increase the absorption of nutrients by the fruit, and promote the size of the fruit [8, 24].

  1. Р. 305-306. It is necessary to explain due to which all types of treatments affect the pH of fruits.

Reply Line 362-371 pH/Acidity explanation was added.

  1. Р. 346-347. The authors of this article indicates the data of other authors on the content of nitrogen in the leaves of plants under different treatments, comparing them with their own data. However, such a comparison is incorrect without indicating the content of these element in the soil. It is better to show how many times the nitrogen content increased in the variants with the treatment of plants compared to the control in your experiments and in the experiments of other authors.

Reply

Increase rates of all nutrient concentrations in T2 -T7 applications compared to control (T1) treatment were calculated and added for nitrogen (Lines 391-392), phosphorus (Lines 413-414), potassium (Lines 428-430), calcium (Lines 435-436), magnesium (Lines 442-443), iron (Lines 449-450), zinc (Lines 455-456), manganese (Lines 459-460), and copper (Lines 465-466).

Reviewer 3 Report

Work that deals with a topic of interest such as the production of strawberries in organic farming with rather original production paths, first of all a single year of experimentation greatly limits the applicability of the protocol followed, the quantities of technical means distributed do not seem to follow a suitable estimation path based on soil characteristics and potential but they seem to have dropped for no particular reason. From the analysis of the soils indicated, the starting conditions seem very severe and confirmed by the results obtained, which are representative of the various theses but practically none capable of justifying these types of production.

The work should be completely revised as a new submission at least by adding another year and carefully motivating doses and choices of technique.

The language is the least aspect of the work, some clarifications in technical terms, but overall sufficient for this version, will be explored in subsequent

Author Response

RESPONSES OF THE REVIEWER 3

Dear reviewer, Dear editor,

Thank you for your comments and suggestions.

Comments and Suggestions for Authors:

Work that deals with a topic of interest such as the production of strawberries in organic farming with rather original production paths, first of all a single year of experimentation greatly limits the applicability of the protocol followed, the quantities of technical means distributed do not seem to follow a suitable estimation path based on soil characteristics and potential but they seem to have dropped for no particular reason. From the analysis of the soils indicated, the starting conditions seem very severe and confirmed by the results obtained, which are representative of the various theses but practically none capable of justifying these types of production. The work should be completely revised as a new submission at least by adding another year and carefully motivating doses and choices of technique.

Comments on the Quality of English Language

The language is the least aspect of the work, some clarifications in technical terms, but overall sufficient for this version, will be explored in subsequent

REPLY: BELOW, OUR ANSWERS WITH THE NECESSARY EXPLANATIONS

We appreciate your detailed response and comments and remain confident that the study's findings and methodology are suitable for publication.

This study is crucial in enriching plant nutrition in organic strawberry cultivation using vermicompost and beneficial microorganisms. It introduces a novel approach to plant nutrition in organic strawberry production. Considering the practical implications of these results, we firmly believe that they should be published yet to benefit organic strawberry producers and the organic fertilizer and microbial-bio-fertilizer sectors.

We acknowledge that the study was conducted under low plastic tunnels, which can be considered a form of partially controlled environmental cultivation. As a result, we confirm that the one-year results from the protected cultivation experiments are indeed suitable for publication.

The soil analysis revealed lower concentrations of available macro and microelements in the experimental soil, as expected in organic agricultural land. The application of vermicompost, PGPR, and AMF effectively dissolved and made these nutrients available for plant uptake while enhancing nitrogen fixation from the air. The limited nutrient availability in the trial soil further underscores the effectiveness of vermicompost-PGPR-AMF applications.

Regarding the dosages of vermicompost, PGPR, and AMF, we adhered to the doses previously determined through rigorous experimentation and preliminary test results, as well as the recommendations provided by commercial manufacturers for strawberry cultivation. Considering these factors, seeking additional dosages or conducting further studies is unreasonable.

We appreciate the opportunity to address your comments and suggestions, and we remain confident in the suitability of our study for publication.

Round 2

Reviewer 3 Report

Paper partially improved but the limits that had previously been highlighted linked to the single year of experimentation remain but above all the protocol on the definition of adequate technical means should be strengthened based on the characteristics and potential of the crop and soil. For example, the considerations added in the review always linked to thesis, soil, doses, productions could be deepened.

English should be perfected especially on technical terms

Author Response

Responses to the Reviewer                                                                                      

Dear Reviewer,

We gratefully accepted the your suggestions and incorporated the necessary revisions into the manuscript. As a result, our article has significantly increased in value. We would like to express our heartfelt appreciaiton to you. Thank you very much.

The additional revisions are highlighted in the manuscript and below in blue color.

Your consideration of the revisions would be greatly appreciated.

Comments and Suggestions for Authors:

Paper partially improved but the limits that had previously been highlighted linked to the single year of experimentation remain but above all the protocol on the definition of adequate technical means should be strengthened based on the characteristics and potential of the crop and soil. For example, the considerations added in the review always linked to thesis, soil, doses, productions could be deepened.

Reply: Below, our additions to the manuscript:

1. Lines 239-242 in blue color was added to the manuscript in the section of 3. Results and Discussion 1. Plant growth parameters

Shoot, root and leaf growth characteristics were significantly affected by the applications and these effects are probably due to the combined effects of vermicompost and biofertilizers. Additionally, factors such as biofertilizer content, dosage, application time and soil characteristics could be be interrelated aspects.

2. Lines 316-318 in blue color was added to the manuscript in section of Results and Discussion 3.2. Total yield and average fruit weight per plant

The strawberry yield results obtained were dependent on the composition, application methods, and dosages of the vermicompost and biofertilizers used, as well as the soil and climatic characteristics of the experimental site.

3. Lines 396-401 in blue color was added to the manuscript in section of Results and Discussion 3.4. Plant nutrient analyses in leaves

One crucial finding of this study demonstrates that relying solely on vermicompost-based fertilization is inferior, and use of complementary top dressing with biofertilization seems to be necessary for superior plant nutrition, especially in T7 conditions. The concentrations of macro and micro nutrients in strawberry leaves could be influenced by the combination of vermicompost and the biofertilizers, their application ratios and timing, and the soil properties employed in the experiment.

4. Lines 535-530 in blue color was added to the manuscript in section of the Conclusion section:

The outcomes of this study depend on several factors, including the species of PGPR and AMF present in the microbial fertilizers, the concentration of viable microorganisms, application rates and timing of the biofertilizers, as well as the physical and chemical properties of the experimental soil in which strawberry plants were organically grown.
